# Mechanical Properties of High Temperature Vulcanized Silicone Rubber Aged in the Natural Environment

**DOI:** 10.3390/polym14204439

**Published:** 2022-10-20

**Authors:** Zhijin Zhang, Jianjie Zhao, Xiaodong Wan, Xingliang Jiang, Jianlin Hu

**Affiliations:** 1Xuefeng Mountain Energy Equipment Safety National Observation and Research Station of Chongqing University, Chongqing 400044, China; 2China Electric Power Research Institute, Beijing 100192, China

**Keywords:** composite insulator shed, mechanical properties, natural aging, nuclear magnetic resonance

## Abstract

Composite insulators operate in harsh field environments all year round. Their various properties and states of aging require attention. It is important to study the performance changes of composite insulator sheds after aging to evaluate the life of insulators operating on grids. For this reason, 22 composite insulator sheds from different factories, with different voltage levels and different ages years were selected to conduct mechanical properties testing. The mechanical properties include hardness, tensile strength, and elongation at break, and were investigated by thermogravimetric (TGA) testing, surface morphology, and nuclear magnetic resonance (NMR) characterization. The changes in mechanical properties of high temperature vulcanization (HTV) composite insulator silicone rubber aged in the natural environment were analyzed, including the reasons for these changes. The results showed that the transverse relaxation time T_2_ of the sample was closely related to its aging state. The more serious the silicone rubber’s aging, the smaller was the T_2_. The state of the composite insulator can be evaluated by using T_2_ and aging years simultaneously. With the actual degree of aging in the silicone rubber intensified, its tensile strength and elongation at break generally showed a downward trend.

## 1. Introduction

Composite insulators have the advantages of negligible mass, high strength, good pollution resistance, and low maintenance workload [1]. Their silicone rubber shed surface is different from other insulators because of its unique hydrophobicity, which considerably improves the anti-pollution flashover capability of composite insulators and is a crucial means to solve the problem of large-scale pollution flashover accidents in China [2]. Composite insulators work in various harsh weather environments all year round and are affected by factors such as ultraviolet radiation, pollution, corona discharge, high humidity, etc. With increasing operating age, the aging problems of insulators are gradually revealed, and the electrical and mechanical properties of insulators decline. This leads to accidents such as sheds cracking and flashover during operation, and these accidents seriously endanger power transmission. Therefore, mechanical performance as an essential property of composite insulators is worthy of attention.

The physical and chemical properties that characterize the electrical performance of composite insulators have been studied worldwide. For example, Mavrikakis et al. [3] tested the aged silicon rubber of composite insulators operating in a coastal salt-spray environment. The results showed that the hydrophobicity of the sheds fluctuated with the appearance and disappearance of the pulverized layer, and the inorganic fillers in the silicone rubber were gradually exposed to the air with the degradation of small molecules at the surface. Chakraborty et al. [4] analyzed thermally aged composite insulators by Fourier transform infrared spectroscopy (FTIR), X-ray photoelectron spectroscopy (XPS), TGA, etc. The study found that with the increase of heat treatment time, the surface of the silicone rubber sheds hardened and faded, the compound SiO_2_ increased, and oxidation occurred. The results of the TGA testing showed that the organic content in the sheds decreased, the inorganic content was exposed, and the proportion of remaining substances in the TGA analysis increased. The remaining substances were inorganic fillers, including Al_2_O_3_ formed after dehydration of aluminum trihydrate (ATH) and silica. Zhang et al. [5] studied the influence of different factors in the natural environment on the hydrophobicity of sheds. Ultraviolet rays and rapid changes in temperature reduced hydrophobicity. The main reasons were the volatilization of small molecular chains from the surface of the insulator and the cracking, oxidation, and free recombination reactions that occurred on the surface of the material. The appearance of a dirt layer and a powdered layer with a certain ratio of ash and salt provided better hydrophobicity for the surface of the sheds [6,7,8,9]. 

Research has also been carried out on the physical and chemical characteristics that affect the mechanical properties of composite insulators. For example, Xie et al. [10] tested the elongation at break and tensile strength of insulator sheds with different operational ages and proposed the idea of damage threshold. They suggested that the tensile strength decreased in stages; before reaching the threshold, the decrease was not obvious, and it decreased linearly after reaching the threshold. Zhang et al. [11] studied mechanical properties of AC-energized silicone rubber in an acidic environment, and the results showed that with the destruction of the surface microstructure, filler silica precipitated under the internal and external concentration variations, lowering tensile strength and elongation at break. 

Ito et al. studied the mechanical and chemical properties of silicone rubber by applying artificial heating, irradiation, and other aging methods. The results show that the performance of the sheds had decreased, and the tensile fracture was mainly elastic deformation [12,13]. Khan et al. studied the effects of inorganic filler ATH on the mechanical properties of sheds. The experiments showed that the particle size and added amount of ATH had little effect on the electrical properties of the sheds, but the mechanical properties were affected [14,15,16]. Wang et al. [17] studied the properties of silicone rubber and fluorosilicone rubber under freezing conditions. In terms of the mechanical properties of silicone rubber, the maximum tensile force and tensile strength of the silicone rubber increased with increasing freezing time. While the hardness of the silicone rubber hardly changed with freezing time, it is evident that low temperature influences the performance of rubber materials.

Internationally, many tests have been carried out on the physical and chemical properties of composite insulators, and the variation rules of some characteristic parameters have been obtained. Due to the small sample size of insulators operating in the field, it is difficult to obtain insulators with different ages in the same environment. Furthermore, the process formulas of different factories are different. The research mechanism is limited to qualitative analysis. In the current study, based on composite insulators located in the test station for different durations of years, the mechanical properties of the sheds were tested for nuclear magnetic resonance, hardness, tensile strength, etc., combined with TGA and microscopic tests, to study the changes in the mechanical properties of composite insulator sheds under the same aging conditions. The research results provide scientific data support for further revealing the mechanical failure mechanism of composite insulator sheds.

## 2. Samples and Test Methods

### 2.1. Test Samples

Composite insulator silicone rubber sheds have aged for different years without electrification at high altitudes in the natural environment of the XueFeng Mountains in Hunan Province. The composite insulators were produced in four different factories, denoted A, B, C, and D. The main components of the composite insulator sheds are methyl vinyl siloxane and inorganic fillers. The proportions of the fillers are different for composite insulators manufactured by different factories, therefore, the initial mechanical properties are different before the aging process begins. To meet the requirements of mechanical and electrical properties including different mechanical loads and creepage distances, different factories have manufacture insulators of different voltage levels. In the study sample there were 22 insulators with five voltage levels, 35 kV, 110 kV, 220 kV, 500 kV, and 800 kV, respectively. Among them, 9 had an aging period of 3.5 years and below, 8 had 3.5 to 8 years, and 5 had 8 years and above. All the composite insulators showed no damage or cracking to the sheds. In the following discussion, “a” indicates units of years of aging.

### 2.2. Mechanical Property Testing and Analysis

#### 2.2.1. NMR Test

NMR tests were carried out to explore the characteristic parameters of the physical and chemical properties of the composite insulator sheds with different operational ages. The NMR sensors used in the experiment are shown in Figure 1.

Before the test, samples were washed with absolute ethanol and dried in the shade for 24 h. Since the upper surface is affected by ultraviolet rays and numerous environmental factors, it can reflect the degree of aging more than the lower surface. NMR experiments were performed on the upper surface of the sample, and the transverse relaxation time T_2_ was obtained. The transverse relaxation time is the time required for the transverse magnetization vector to decay to 37% of the maximum value [18]. For a given substance, the size of T_2_ can reflect the aging degree of the sample, the lower the T_2_, the greater the aging degree of the sample [19]. Room temperature and sample temperature affect the determination of T_2_, and the magnetic field’s polarization causes the sample’s heating. The experimental data were measured at constant room temperature in a single session. In the NMR experiment, the CPMG (Carr–Purcell–Meiboom–Gill) pulse sequence was utilized to excite the sample, as shown in Figure 2; d is the pulse width, T_echo_ is the echo time, A_180__°_ and A_90__°_ are the 180° pulse amplitude and the 90° pulse amplitude, respectively.

The CPMG pulse parameters in the experiment were set as in Table 1.

#### 2.2.2. Hardness Test

An electronic shore hardness tester was utilized to measure the hardness of the composite insulator sheds. The test method corresponds with standard ISO 48-4 [20]. The same sample was tested from the inner side of the shed to the outer and the median of the test values was taken as the sample hardness value. During the test, the indenter was placed perpendicular to the sample surface, the pressure foot was perpendicular to the sample surface, and the pressing speed was not more than 3.2 mm/s. The spring test force was maintained for 3 s before reading.

#### 2.2.3. Mechanical Property Test

These tests were conducted according to the ISO 37 standard guidelines [21]. The sample was cut into a dumbbell shape with a JZ-6010 punching machine. The dimensions are shown in Figure 3. The total length A was 35 mm, the experimental length C was 10 mm, the width D of the narrow part was 2 mm, and the width B of the end was 6 mm. To better control the longitudinal thickness variation of the cut samples, the cutting direction was parallel to the tangent on the outer circumference of the insulator.

The test used AGX with a desktop electronic universal testing machine produced by SHIMADZU company, as shown in Figure 4. In this study, the load capacity of the load sensor was selected as 5 kN, and TRAPEZIUMX software was employed to monitor the elongation and tensile force when the test sample was broken.

Before the test, the thickness of the C section of the experimental length of the sample was measured. When measuring, the cross-section of the test sample was uniformly stressed, and the gripper movement speed was set to 500 mm/min. During the test, if the sample did not break in the narrow part, the result was discarded and another sample was retested. Each sample was measured three times, the tensile strength was calculated separately, and the final results were averaged. The tensile strength *T_s_* can be calculated according to Formula (1), and the elongation at break *E_b_* can be calculated according to Formula (2).
(1)$$T_{s}=\frac{F_{m}}{D\cdot t}$$
(2)$$E_{b}=\frac{L_{b}-L_{0}}{L_{0}}\times 100\%$$
where *T_s_* is the tensile strength (MPa), *F_m_* is the tensile force at break (N), *D* is the width of the narrow part of the sample (mm), *t* is the thickness of the sample (mm), *E_b_* is the elongation at break (%), *L*_0_ is the initial test length of the sample (mm), *L_b_* is the test length of the sample at the moment of rupture (mm). 

#### 2.2.4. TGA Analysis

TGA is a simple technique for measuring change in weight with temperature. It provides decisive data for material and product design as well as information about aging stability. The TGA analyzer model used in this study was TGA/DSC1/1600LF. The TGA analyzer is shown in Figure 5. In this study, the heating rate was 10 °C/min, the carrier gas was nitrogen (50 mL/min), and the scanning range was from room temperature to 750 °C.

#### 2.2.5. Surface Microscopic Topography Test

In this work, an environmental scanning electron microscope (SEM) was utilized to observe the micro-morphology of the composite insulator silicone rubber shed sample, which was viewed from all angles.

The environmental scanning electron microscope was produced by Thermo Fisher. The accelerating voltage was 0.2~30 kV, the resolution 1 nm (30 kV)~3 nm (1 kV), and the magnification 6~2,500,000. Due to the insulating nature of silicone rubber, all the specimens were sputter-coated with a thin layer (15 nm) of gold.

## 3. Results and Discussion

### 3.1. NMR Test Results

The main material of the composite insulator shed samples was polydimethylsiloxane (PDMS). When the silicone rubber ages on the composite insulator in the natural environment, ultraviolet rays break the chemical bonds with lower bond energy in the silicone rubber, such as Si-C bonds and C-H bonds. With further aging, the Si-O bond energy on the main chain decays, and the bond energy between the Si atom and the methyl or vinyl groups on the side chain changes accordingly, thereby affecting the H atom in the groups. The increase in cross-link density also reduces the mobility of H-containing groups. The quantity of H atoms in PDMS is high, and a large signal intensity can be obtained by using H as the resonance object. Nuclear magnetic resonance technology can reveal the H content and state in the silicone rubber by using the resonance characteristics of the H atom, to reflect the degree of aging.

The samples were tested by NMR according to the method described above. The CPMG echo signal image obtained from a particular test is shown in Figure 6, and the NMR signal image is shown in Figure 7. T_2_ is the transverse relaxation time.

The years of aging of the samples from each factory and the transverse relaxation time T_2_ are shown in Figure 8.

From the results in Figure 8, it can be seen that:(1)With the increase in years of aging, the T_2_ values of the composite insulator sheds decreased, and the two showed certain linearity. For example, the 35 kV sheds from factory A had aging periods of 1, 3.5, and 8 years, and their T_2_ values were 122.5 ms, 120.6 ms, and 118.7 ms, respectively, indicating a gradual decrease.(2)The sheds’ T_2_ values were different at different voltage grades with the same aging periods from the same manufacturer. For example, factory A produced 35 kV and 110 kV sheds and with aging periods of 1 year their T_2_ values were 122.5 ms and 114.8 ms, respectively. The 110 kV and 220 kV sheds from factory C with aging periods of 6 years had T_2_ values of 111.7 ms and 122.1 ms, respectively.(3)Sheds with the same aging period and the same voltage level from different factories had different T_2_ values. For example, factories B and C both produced 220 kV sheds which had an aging period of 6 years, with T_2_ values of 115.0 ms and 122.1 ms, respectively. Factories A, C, and D all made 110 kV sheds with aging periods of 6 years, and the T_2_ values of the three were 102.5 ms, 116.3 ms, and 125.0 ms, respectively.(4)The aging speeds of sheds with the same voltage levels from different factories were different, as reflected in the linear fitting line showing different slopes. For example, the slopes of the fitting lines of the 110 kV sheds from factories A and C were 2.46 and 1.63, respectively. The greater the slope, the faster the T_2_ value decreased with the aging period, i.e., the faster the aging speed.(5)There were also differences in the overall T_2_ values of sheds from different factories. The T_2_ value of sheds from factory D maintained a high level during the aging period, and the T_2_ of the sample aged for 10 years was 123.2 ms. The maximum value of T_2_ of the aged samples from factory B was 115.8 ms, and the aging period was 3.5 years. For factory C that value was 122.1 ms, and the aging period was 6 years.

There was a positive correlation between years of aging and the transverse relaxation time T_2_ of natural aged composite insulator sheds, and T_2_ gradually decreased with the increase in years of aging. The different insulator formulas and production processes of factories are reasons for the differences in T_2_ value and rate of decline. Therefore, the recorded years of aging cannot be used as the only reference standard. The transverse relaxation time T_2_ obtained from the tests and the recorded years of aging should be comprehensively considered to analyze the actual aging of samples, and the influence of different factories and voltage levels should also be considered.

### 3.2. Hardness Test Results

The samples were tested for hardness according to the method described above. The details are shown in Figure 9.

From the results in Figure 9, it can be seen that:(1)With the increase in years of aging, the hardness of composite insulator sheds generally showed an increasing trend. For example, the aging periods of the 35 kV sheds from factory A were 1, 3.5, and 8 years, and the hardness was 68.8, 70.8, and 78, respectively. The 110 kV samples from factory D showed an upward trend in general from 3.5 years to 10 years, and the hardness increased by 2.61%. The hardness of the samples from factory D with an aging period from 3.5 years to 6 years and the hardness of the 220 kV sheds from factory C decreased slightly. Combined with the high overall value of T_2_, the overall chemical structure did not change dramatically, and the actual degree of aging was not high. It can be speculated that the drop point may have been caused by different onsite sampling locations, according to the sheds’ leeward aspect, exposure of inner and outer positions to ultraviolet rays, etc. The severity of natural aging factors differed, resulting in diverse test results for the same sheds taken from different sampling positions.(2)Sheds with the same years of aging and voltage levels from different factories had different hardness values. For example, factories A, C, and D all produced samples with voltage of 110 kV and an aging period of 6 years, with hardness 75.2, 83, and 76.5, respectively.(3)There were also differences in the overall hardness of the sheds from different factories. The hardness variation range of the sheds from factories B and D during the aging period was 2, and the ranges for A and C were 9.8 and 6, respectively. The effect of ultraviolet aging on the hardness of samples from factories A and C was more obvious.

In general, the hardness of sheds increased with the increase in years of aging. Research analysis has indicated that, in the early stages of the aging period, ultraviolet light carrying photon energy severs the Si-C bonds and C-H bonds with lower bond energy in the sheds’ polymer material [22]. Breaking the polymer link plays a leading role in the aging process, and the overall structure becomes loose. As the aging time increases, the aging degree deepens, the cross-linking reaction becomes larger, the chemical bonds are broken and recombined to form a Si-C-Si structure, and Si precipitation increases the silicone rubber’s surface hardness. Under the comprehensive action of the above factors, the surface hardness of the silicone rubber increases.

As mentioned above, the actual aging of the samples was be analyzed with consideration of T_2_ and the recorded aging years. For example, data for the 500 kV sheds from factory B reflects the superiority of adding T_2_ as an aging index; the hardness of the sheds with aging time of 1 year and 10 years was 80.5 and 81.5, respectively. Although the years of aging varied, the hardness hardly changed, and T_2_ of the two were 114.1 ms and 113.7 ms, respectively. It can be seen from the NMR results that their actual degree of aging was almost the same, and the degree of cross-linking was equivalent.

### 3.3. Mechanical Properties Test Results

The mechanical properties were tested according to the above method, and the relationship between the mechanical properties of the samples from each factory and the years of natural aging was analyzed and compared. The results are shown in Figure 10.

From the results in Figure 10, it can be seen that:(1)With the increase in years of aging, the tensile strength of composite insulator sheds generally showed a downward trend. For example, the years of aging of the 35 kV sheds from factory A were 1, 3.5, and 8 years, and the tensile strengths were 4.58 MPa, 3.88 MPa, and 3.79 MPa, respectively. For sheds from factory D, the tensile strength dropped from 6.77 MPa to 6.63 MPa with increased aging from 3.5 to 10 years.(2)The tensile strength was different in sheds of different voltage levels but from the same factory with the same aging period. For example, the 35 kV and 110 kV sheds aged for one year in factory A had tensile strength values of 4.58 MPa and 4.41 MPa, respectively. The 220 kV and 110 kV sheds aged 6 years from factory C had tensile strength values of 5.82 MPa and 4.89 MPa, respectively.(3)Sheds with the same aging period and voltage level from different factories had different tensile strengths. For example, the factories A, C, and D all produced samples with a voltage level of 110 kV and an aging period of 6 years, and their tensile strengths were 3.94 MPa, 4.89 MPa, and 6.44 MPa, respectively.(4)Sheds from different factories showed the same phenomenon of rapid performance decline in the early stages of aging. The tensile strength of the 35 kV sheds from factory A decreased by 15.2% and 2.3%, respectively, in the aging period of 1 year to 3.5 years and of 3.5 years to 8 years. The tensile strength of the sheds decreased faster when initially aged by natural factors. This phenomenon is reflected by the changing slope of the straight line in the figure. This phenomenon was also apparent in the 110 kV factory C and 220 kV factory B samples.(5)There were similarities and differences in the overall tensile strengths of sheds from different factories. For example, the measured minimum tensile strength of the sheds from factory D after aging is 6.44 MPa, which was much higher than the other three factories. The tensile strength of the sheds from each of the four factories showed the same phenomenon, generally higher at high voltage levels than at low voltage levels.

In general, with the increase in years of aging, the overall tensile strength of the sheds decreased. It can be speculated that where the curve for factory D declines and then rises after aging, the effect may be caused by different sampling locations on site. With the increase in years of aging, the T_2_ values of the three samples fluctuated by no more than 2.57%. Compared with the NMR results of the other samples, this group of samples maintained a very high T_2_ value. That is, in this natural environment, the degree of aging was not obvious in terms of increasing mechanical properties.

DL/T 376-2010 [23] stipulates that the mechanical tensile strength of shed materials for insulators should not be less than 4.0 MPa, and the elongation at break should not be less than 150%. Among the four factories, only the samples from factory D continued to show excellent tensile strength after 10 years of natural aging, and those sheds had better overall aging resistance in terms of mechanical properties in the high-altitude ultraviolet environment. The mechanical properties of samples from factory C were slightly worse than those from factory D, while the mechanical properties of the 35 kV level samples from factory A and the 220 kV level samples from factory B declined rapidly in this environment.

The relationships between the tensile strength, elongation at break, and T_2_ (lower than the specified value not considered) are shown in Figure 11 and Figure 12 below.

From the fitting results in Figure 11, there were seven samples whose tensile strength was less than the specified 4.0 MPa after aging, and they are marked with red symbols. For the remaining samples, they are marked with black symbols, as the actual aging degree intensified, T_2_ decreased, the tensile strength generally showed a downward trend, and T_2_ was positively correlated with the tensile strength.

Analysis shows that under the irradiation of ultraviolet energy, the chemical bonds with lower bond energy are separated, manifesting as the decline of mechanical properties including tensile strength. With the increase in years of aging, the number of broken chemical bonds increases, the microscopic cross-linking of free radicals occurs, and the micro-dynamic process of fracture and cross-linking is reflected in an overall decrease in tensile strength at the macro level. With the increase in years of aging, the functional group components inside the silicone rubber and the elemental contents change. The backbone integrity is destroyed after aging, and the polarity degree increases. Its overall performance exhibits a decrease of mechanical properties. 

According to the results in Figure 12, it can be seen that three samples from factory B did not meet the 150% elongation at break specified in the standard above, and they are marked with red symbols. Meanwhile, the samples from other factories maintained good elongation at break during aging, and they are marked with black symbols. With the decrease of T_2_, the samples’ overall elongation at break showed a decreasing trend.

The above results show that the years of aging and T_2_ were consistent when characterizing the mechanical properties of the samples. These two characteristic parameters should be combined during the monitoring and analysis of composite insulator sheds during aging.

### 3.4. Thermogravimetric (TGA) Analysis Results

The main components of composite insulator sheds are methyl vinyl siloxane (base glue) and inorganic fillers. The inorganic fillers are generally aluminum trihydrate (ATH) and silica. As an important additive, ATH can improve flame retardancy, electrical tracking resistance, corrosion resistance, electrical properties, and UV aging resistance of silicone rubber. The composite insulators manufactured by different factories contained different proportions of each component, some may add more inorganic filler in order to save costs, and the shed will show different resistance during the aging process. According to the above analysis, there were numerical differences in the test results of samples after aging from different manufacturers, and they followed the same pattern of change. Taking samples from factory C as an example, the TGA analysis curve of the samples in the 110 kV group is shown in Figure 13, and the sample data is shown in Table 2.

It can be seen from the curves and graphs that different years of uncharged aging had an impact on the thermal gravimetric test results for the sheds. With the increase in years of aging, T_2_ became smaller, the proportion of weight loss in the first and second stages decreased, and the proportion of remaining mass increased.

The specific TGA analysis data are shown in Table 3.

According to the analysis, the first stage of weight loss was mainly due to the dehydration of ATH in the inorganic filler of the sheds. This dehydration of the ATH is stated as:$$2\left[\mathrm{Al}{\left(\mathrm{OH}\right)}_{3}\right]{=\mathrm{Al}}_{2}O_{3}{+3H}_{2}O$$

The more serious the aging, the more inorganic fillers are precipitated from the pores caused by the aging, and the decomposable amount after heating decreases. The weight loss in the second stage was mainly the thermal decomposition of PDMS. Similar to the previous stage, the more serious the aging, the less PDMS available for pyrolysis. Decomposition of PDMS may be caused by the release of the siloxane side chain (CH_3_) from combustion, as shown in the following (Figure 1):

The remaining substances are inorganic fillers, including Al_2_O_3_ formed after dehydration of ATH and silica.

ATH in sheds has a spacing effect. The addition of ATH can limit the re-cross-linking of the broken chemical bonds of the silicone rubber, and the inorganic fillers are not affected by ultraviolet rays, so the more fillers, the less the shed surface is affected by radiation. Aging leads to the appearance of voids, and the inorganic fillers are exposed to the air and precipitate [24]. The reduction of inorganic fillers reduces the blocking points in the base rubber, but at the same time, it makes the internal cross-linking of the matrix easier and the effect of ultraviolet radiation on the sheds becomes greater. It is a process of micro-dynamic confrontation.

The above results show a correlation between the precipitation of inorganic fillers inside the silicone rubber shed and the decomposition of the base rubber under uncharged aging, with ultraviolet rays as the main aging factor. With the increase in years of aging, the value of T_2_ decreases, the content of base rubber in the sheds gradually decreases, and the content of inorganic filler increases, which is consistent with the gradual decrease of tensile strength of C-1, C-2, and C-3.

### 3.5. SEM Test Results 

The samples from factory C are taken as examples. The SEM test images of the samples of the 110 kV group are shown in Figure 14.

In Figure 14, (a) refers to C-1, (b,c) is C-2, and (d) is C-3. No obvious defects were observed unaided on the surface of sample C-1, but it was seen from the SEM image that cracks and holes began to appear on the surface of the sheds, and there were no obvious rough protrusions or deep pits. There were obvious tortoiseshell cracks resembling the shell of a turtle on the surface of sample C-2, which were not observed on the surfaces of other samples in this group. Microcracks on the material cause local stress concentration. Since the depth of the NMR test was in millimeters, it can be speculated that the appearance of tortoiseshell cracks was a vital reason why the tensile strength and transverse relaxation time T_2_ of C-2 were smaller than those of C-1. At the same time, the polymer materials in the cracks were directly exposed to the natural environment, and the degree of intermolecular cross-linking became larger, leading to further aging. The surface hardness of silicone rubber is closely related to the degree of intermolecular cross-linking; specifically, hardness increases with greater cross-linking density [25].

With the increase in years of aging, deep pits of different sizes were observed on the surface of the C-3 sample, and the surface of the silicone rubber gradually changed from flat to rough, and holes appeared. After the number of holes increased, water easily entered the material, causing decreases in the mechanical properties of the silicone rubber.

The bonding effect of inorganic fillers and silicone rubber matrix materials is not ideal, and an increase of filler ratio will reduce the bonding degree of the base rubber body, the mechanical strength, and the tensile strength [26]. The elongation at break can also be affected by the content of inorganic fillers and base rubber [27]. Silicone rubber itself has good stretchability and elasticity. The addition of inorganic fillers introduces defect-like blocking points in the silicone rubber matrix, which reduces the elongation at break.

## 4. Conclusions

Based on NMR technology and mechanical properties testing, this paper has taken the natural aged composite insulator shed as the research object, systematically analyzed and studied its mechanical properties after aging, and here draws the following conclusions:Natural aging reduces the mechanical properties of silicone rubber insulator sheds. The main effect is that with the increase in years of aging, the surface of the material becomes harder, and the tensile strength decreases. The elongation at break value fluctuates, but generally shows a decreasing trend.There was a certain linearity between the transverse relaxation time T_2_ of the sheds, measured by NMR testing, and the recorded years of aging, showing that the longer the aging period, the smaller was the T_2_. There was consistency in measuring the mechanical properties of sheds with years of aging and T_2_ as characteristic values.Natural aging is mainly caused by environmental factors. The samples in this paper were mainly affected by snow, fog, and ultraviolet rays, from which ultraviolet energy damages the chemical bonds of the silicone rubber sheds. Microscopically, this process is a dynamic confrontation between chemical bond breaking and cross-linking. Macroscopically, the surface morphology of the sheds and the content of inorganic fillers are changed, thereby affecting the mechanical properties.

## Data Availability

Data will be made available on reasonable request.

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
