# Peer review of "Mechanical Properties of High Temperature Vulcanized Silicone Rubber Aged in the Natural Environment"

_polymers, 2022, doi:10.3390/polym14204439_

Round 1
Reviewer 1 Report
L20-22 ...With the deepening of the actual....I could not understand this sentence in the abstract.
L26-27 The authors need to change the references number to arabic ones throughout the text.
L33 ...and their electrical and mechanical properties have declined. This sentence is very difficult to comprehend.
L35 which seriously endangers "the" power transmission.
L36 I would put a final point before "therefore". Very long sentence
L39 ...are studied at home and abroad. I could not understand this sentence in the introduction.
L44 R. Chakraborty et al... Here, the author should put next the reference iv
L47 It is very difficult to understand the sentence part: ... the Si element compound state increased. The author must indicate to which compound he is referring
L49 What are the remaining substances in the TGA?.
L50-51 The rapid changes in temperature and ultraviolet rays will reduce hydrophobicity. The authors can provide an explanation to this phenomenon from a chemical point of view using mechanisms of thermal and photo-oxidative degradation.
L52 For me, the appearance of a dirt layer and a powdered layer with a certain ratio of ash and salt can provide better "hydrophilicity" and not hydrophobicity for the surface of the sheds. The presence of salt and ash produced on the surface enhances conductivity together hydrophilicity.
L57 They think the tensile strength decreases in stages. The word "that" is missing. You mean "they think that the tensile ...
L61-62 There are two ideas in this sentence that should be separated by "and", performance.... and tensile fracture. Thus, it is not clear what the authors are referring to.
L64-65 I do not understand the sentence. You mean "The experiments show ..... and added amount of ATH ..."
L67-69. What do you mean by this sentence? The sentence is incomplete using "although"
L99 You mean "T2 relaxation time", this term is better known in NMR
L101 You mean "the lower the T2, the greater the aging degree of the sample".
L131 I do not understand the "stroke" word in this sentence. You mean "...to monitor the elongation and tensile force when the test sample breaks (Eq. 1 and 2)"
L137 "...the gripper movement speed should be set to 500 mm/min"
L139 "....retested"
L161 "In this work..."
L162 You must to state what is SEM (scanning electron microscope) and then use these acronyms in L164
L169 You mean "When the silicone rubber on composite insulators ages in the natural environment". Please rephrase this sentence
L170-172 "...will cut off the chemical bonds with lower bond energy in the silicone rubber ... state of H atoms in the material will change accordingly" ?. This sentence in the chemical word is not comprehended.
First, there is a rupture of chemical bonds, it is not "cut off". Second, what do you mean by state of H atoms?. Please if you refer at radical atoms due to the degradation, it should be more clearly stated to chemistry readers.
L172 "...The increase in link density" You must say The increase in cross-linking density
L173 The main material of the composite insulator shed is polydimethylsiloxane (PDMS). This sentence should be put before, e.g. in L169 at the beginning.
L174 You mean "amount of H atoms in PDMS is high", not H content. This sounds like it's not going to be read by chemists.
L185-186 The authors conclude that there is certain linearity in Figure 8 with two and three points. Mathematically, two points is the minimum amount of points to establish a straight line. The authors should have more points to reach that conclusion. There are few experimental points per factory
In the Figure 8 there are several factories: a, b, c, and d. What is the difference between them? This point should be clearly stated in a table, showing the difference between factories. I was not able to finish reading. By the way, you stated that a factory contained samples of 800 KV and no results were presented.
In general terms, the results are qualitatively described, but they are not chemically justified. Why do the results differ between factories at the same voltage?
The authors do not discuss loss of solvent, which is used as plasticizer in the manufacturing of insulator, with the increase of aging years. This fact also causes hardening of material, not only is the rupture of chemical bonds.
By the way, the authors might have provided crosslinking density of insulator silicone rubber sheds with the aging years. T2 relaxation in RMN permit to measure such density (see Kautschuk und Gummi Kunststoffe 55(9):460-463 Real-time 1H NMR relaxation study of EPDM vulcanization).
Conclusions in this manuscript are not a novelty. From a chemical point of view, these conclusions are expected having a minimum knowledge in polymers.
Reviewer 2 Report
This article is interesting however presents limitations in both introduction and discussion sections.
Keywords should be written according to mesh terms and alphabetical order.
You should write the full name before abbreviations there are a lot to mention and especially the introduction section
Lack of sample size calculation
Discussion should be supported by novel articles
Please add limitations for the study
References need further checking. You should add more references.
A lot of grammatical errors. Please correct.
Reviewer 3 Report
The study has good scientific merit but extensive modifications are necessary.
Please revise the title to better reflect the content of the study.
Line 67-69 – Hanging sentence
The methodology method has to be revised significantly. The whole experiment is not clearly described.
Line 81 – how many years of aging?
Line 83 – “The composite insulators come from four different factories, denoted A, B, C, and D.” more information is needed, what is the difference between this composite?
Line 85-86 – what do you mean by 3.5a and below, 3.5a to 8a, and 5 with 8a and above? What is a?
Line 105 – please define CPMG
Line 120 – Mechanical property – is it only tensile strength was tested? If so, please change the title of the subsection
Line 183 and Figure 8, please subscript T2. Why composite insulator from different factories having different voltages? This should be described in details in Methodology section.
Line 225 – “The 110kV samples of factory D showed an upward trend in general, from 3.5 years to 10 years” How about the 6-year aging? It decreases. Should it be addressed?
Section 3.2 and 3.3 – the results are very well-presented. However, there are lacking of discussion on why such findings were observed? The main concern here is the justification of samples from different factories and having different voltages? What make the difference between them and the resulted observation? The authors should discuss more from this point of view.
Line 340 – “Taking samples from factory C as an example”, what is the particular reason for choosing samples from factory C?
Section 3.4 – more discussion on the TGA should be added. Explain on what causes such observation.
Line 378 – what is tortoise cracks?
Line 392-397 – it this relevant? Did the authors conduct research on content of inorganic fillers and base rubber?
Round 2
Reviewer 1 Report
The authors made major revisions. Now, this manuscript can be published
Reviewer 3 Report
The authors did a great job in addressing all the comments raised. I think it is acceptable now.